# Description and Preliminary Simulations with the Italian Vineyard Integrated Numerical Model for Estimating Physiological Values (IVINE)

**Valentina Andreoli [1,\*], Claudio Cassardo [2,3], Tiziana La Iacona [4] and Federico Spanna [4]**

1   Department of Physics, University of Torino "Alma Universitas Taurinorum", 10125 Torino, Italy
2   Department of Physics and NatRisk Center, University of Torino "Alma Universitas Taurinorum", 10125 Torino, Italy; claudio.cassardo@unito.it
3   College of Environmental Science and Engineering, Ewha Womans University, Seoul 03760, Korea
4   Phytosanitary Sector, Regione Piemonte, 10144 Torino, Italy; federico.spanna@regione.piemonte.it (T.L.I.); tiziana.laiacona@mail.regione.piemonte.it (F.S.)
\*   Correspondence: valentina.andreoli@unito.it; Tel.: +39-011-670-7438

**Abstract:** The numerical crop growth model Italian Vineyard Integrated Numerical model for Estimating physiological values (IVINE) was developed in order to evaluate environmental forcing effects on vine growth. The IVINE model simulates vine growth processes with parameterizations, allowing the understanding of plant conditions at a vineyard scale. It requires a set of meteorology data and soil water status as boundary conditions. The primary model outputs are main phenological stages, leaf development, yield, and sugar concentration. The model requires setting some variety information depending on the cultivar: At present, IVINE is optimized for *Vitis vinifera* L. Nebbiolo, a variety grown mostly in the Piedmont region (northwestern Italy). In order to evaluate the model accuracy, IVINE was validated using experimental observations gathered in Piedmontese vineyards, showing performances similar or slightly better than those of other widely used crop models. The results of a sensitivity analysis performed to highlight the effects of the variations of air temperature and soil water potential input variables on IVINE outputs showed that most phenological stages anticipated with increasing temperatures, while berry sugar content saturated at about 25.5 °Bx. Long-term (60 years, in the period 1950–2009) simulations performed over a Piedmontese subregion showed statistically significant variations of most IVINE output variables, with larger time trend slopes referring to the most recent 30-year period (1980–2009), thus confirming that ongoing climate change started influencing Piedmontese vineyards in 1980.

**Keywords:** viticulture; crop model; phenology; physiological processes; climate; micrometeorology; microclimate; climate change

## 1. Introduction

Grapevines are strongly dependent on environmental conditions, and several factors can influence their quality and productivity: Weather, climate, soil fertility, and management practices, among others. An increase in temperature has an important impact on crop growth and yield [1].

There is an increasing interest in the use of crop growth models as tools to assess climate variability and change in crop yields and quality [2–4]. Crop growth models can help to evaluate interactions between cultivar, the environment, and management strategies, and provide an instrument to understand complex plant processes and how they are influenced by pedoclimatic and management conditions. Crop growth models are currently employed at a regional scale for agricultural (yield or

quality assessments) or environmental applications (crop water requirements, nitrate leaching) and as a tool to support the process of decision-making and planning in agriculture [5–9].

In particular, the study described in Costa et al. [10], related to the application of crop modeling to Portuguese viticulture, provided a review of research on grapevine models as key decision-supporting systems under current and future climatic conditions.

Many crop growth models operate on a daily time step and simulate the evolution of variables of agronomic interest through daily accumulation. Weather conditions are the input data that drive the crop models and have a noticeable effect on yield and other model outputs. Thus, these kinds of data need to be described accurately.

Since 1960, the Wageningen group has developed crop growth models of varied degrees of complexity for different purposes [11]. For example, the generic model BACROS was developed and improved between the 1960s and 1970s: This modeling approach was used by the modeling group [12,13], while the generic crop model SUCROS, developed in the 1980s [14], represented the basis of most recent Wageningen group crop models, such as WOFOST, ORYZA, INTERCOM, and LINTUL [15–17].

Some crop growth models are adaptable to various crops and can simulate crop growth and plant development, as well as water and nitrogen balances: This is the case of some Wageningen models, as well as STICS [18,19], developed since 1995 at the French National Institute for Agricultural Research (INRA). The STICS model is driven by daily climatic data and simulates crop growth, soil water, and nitrogen balance. It is adaptable to various crops by the use of generic parameters relevant for most crops and by the introduction of physiology and management formalization, chosen for each crop.

Specific crop growth models have also been developed to simulate grapevine growth and development. Among all models, we can mention a simple model for the simulation of growth and yield of a grapevine, specifically the Sangiovese vine [20]; the source-sink model developed to simulate the seasonal carbon supply and partition among reproductive and vegetative parts of a vine [21]; a model predicting phenology, leaf area development, and yield [22], and finally a decision-supporting system for sustainable management of vineyards and real-time monitoring [9]. Furthermore, a model for predicting daily carbon balance and dry matter accumulation in grapevines has been implemented [23].

In addition, the biophysical grape berry growth module described in Reference [24] has been developed and integrated with the whole-plant functional–structural model GrapevineXL and calibrated on two famous international varieties.

The generic crop model STICS has been adapted for grapevines and evaluated for different vineyards and cultivars in France [25]. Its ability to represent phenology, biomass production, yield, and soil water content has been studied for Portuguese grapevines and vineyards located in Chile and France [26,27].

Generally, crop growth models include specific modules calculating the occurrence of phenological stages that can also be used as stand-alone routines. Several models predicting the bud-burst date of a grapevine have been tested and compared [28]: The results of this study showed that calculation of dormancy break, provided by the BRIN model, is not a critical factor for improving the prediction of a bud-burst date under current climatic conditions, but it could become important in future climates. Models simulating the timing of flowering and veraison of grapevines have been tested, and a general phenological model (the spring warming model named the Grapevine Flowering Veraison model (GFV)) was developed and optimized [29], showing the best results in predicting flowering and veraison dates for different varieties.

Finally, grapevine phenology has recently been studied in connection with climate change by means of grape harvest dates used to reconstruct past climate [30] and phenological data of different cultivars in the Veneto region from a long-term collection [31]. The results showed that models used to relate temperature to grape harvest dates can be accurate, but both types of methodologies

(linear regression and process-based phenological models) can induce some biases in temperature reconstruction. Grapevine phenology was influenced by the observed warming in the Veneto region: Flowering, veraison, and harvest dates were anticipated during the examined period (1964–2009).

From this overview, it is clear that there have been several studies on crop modeling applied to vineyards. However, few models have been specifically developed for studying grapevines [32], often deepening only certain aspects of crop growth, and few of them could evaluate water balance in vineyards. For this reason, we decided to develop a new crop model instead of adapting and implementing other existing models. The aim of this paper is to present the crop growth model Italian Vineyards Integrated Numerical model for Estimating physiological values (IVINE) [33,34]. IVINE is able to simulate a wide set of phenological and physiological parameters for vineyards using physically based equations for processes such as water balance and photosynthesis, and empirical equations for others. Since our intention was to study *Vitis vinifera* L. Nebbiolo, of which there are few studies in the literature and very few applications using crop models (none of them complete), we calibrated IVINE for cv. Nebbiolo. This cultivar usually is characterized by a large interval of time between the flowering and harvest stages, larger than for other more widespread and studied cultivars. Thus, the model calibration required particular attention.

Here, we would like to develop a grapevine growth model, based on previously described methods, studying the effects of climate change on phenology and yield in the northwestern Italian region of Piedmont. In fact, crop models can be applied to study vineyard complex agroecosystems and multilevel environments. However, before examining the consequences of future climate change on the vineyard environment, it is necessary to verify if and how much the selected crop model is able to provide an adequate representation of these processes in the present and recent climatic conditions.

The paper, after the model description, contains three sections. The first one is dedicated to IVINE validation with field observations. The second one presents a sensitivity analysis on the most important variables (air temperature and water potential) among the IVINE inputs (which also include air relative humidity, wind speed, global solar radiation, photosynthetically active radiation, atmospheric pressure, and soil temperature). The third describes long-term simulations (60 years) carried out over a specific wine area in the northwestern Italian region of Piedmont (Langhe, Roero, and Monferrato), famous for cv. Nebbiolo.

## 2. Materials and Methods

### 2.1. The IVINE Model

The numerical model IVINE is a crop growth model created to simulate physiological and phenological vineyard conditions. The model requires a set of meteorological data and vineyard and soil information. It runs on daily steps, and phenological phases dictate the timing of different model routines.

The required boundary conditions, provided during the simulation, are hourly data: Air temperature and relative humidity, solar global radiation, photosynthetically active radiation, soil temperature, soil water content, wind speed and direction, and atmospheric pressure.

Other data required as inputs (about vineyard and soil characteristics) are geographic information (latitude, longitude, and elevation), soil hydrology, variety characteristics, and vineyard management information. Soil parameters (the b-power parameter [35], hydraulic conductivity, soil porosity, wilting point, field capacity, saturated soil water potential, and soil thermal capacity) are required and can be evaluated according to empirical equations [35] by means of organic matter and sand and clay soil percentages, if available, or according to the U.S. Department of Agriculture soil textural classes [36–38]. We are aware of the approximations introduced by such kind of parameterizations, but we think that an even larger error could be produced by the large variations in soil parameters within the same soil class. Unfortunately, in the absence of specific measurements at a local scale, we think that this kind of error cannot be reduced. The presence of a steep slope on terrain has [32] a direct effect on air

temperature, solar radiation, and soil status (temperature and moisture [39]). IVINE does not consider explicitly this parameter in its equations, but terrain slope information can be implicitly given to IVINE by selecting an accurate set of boundary conditions.

The IVINE model also requires the setting of some experimental parameters that depend on the cultivar (plant density, thermal thresholds, sugar content threshold at harvest, mean number of clusters per plant, and mean number of berries per cluster) and the site (soil layers number, texture, and depth). The following data about vineyard management are also required: The date and the severity of trimming and thinning (in case they are not available, IVINE prescribes fixed values at fixed dates). At present, the model is optimized for cv. Nebbiolo, since in Piedmont the most famous wines are produced from this cultivar.

The main model outputs are timing of the main phenological stages, leaf development, yield, and berry sugar concentration.

The occurrence of main simulated phenological stages (expressed in Julian days (JDs), used instead of calendar dates to represent the latter by integer values, starting from 1 on January 1st and ending with 365 or 366 on December 31st, and restarting the count at the beginning of each year) are dormancy break, bud-burst, flowering, fruit-set, beginning of ripening, veraison, and harvest: Their simulations use some thermal thresholds and the berry sugar concentration.

The phenological phase of dormancy break is simulated by means of chilling units ($Cu$) [28,40,41]: Its calculation starts on August 1st (a date close to the period in which the highest annual temperature is usually observed), and the phenological stage occurs when a critical amount of chilling units (100 $Cu$) is reached. Chilling units (Equation (1)) are calculated by means of maximum and minimum daily temperatures ($T_x$ and $T_n$) and a parameter $Q$, set equal to 2.17 [28], while $n$ refers to days [28]:

$$Cu = Q^{-Tx(n)/10} + Q^{-Tn(n)/10} \qquad (1)$$

The postdormancy time period is calculated from the dormancy break using a sum of hourly temperatures $T_r(h,n)$, called growing degree hours or $GDH$, defined in Equation (2) and obtained by the method of Richardson [42,43], used in the BRIN model [28]. If not available, hourly temperatures are derived as in Section S1. The calculation stops when a threshold value equal to 8050 $GDH$ (derived from the IVINE calibration) is reached:

$$GDH = \sum_h T_r(h, n) \qquad (2)$$

The phenological phase of flowering (fruit-set) is simulated by means of growing degree-days $GDD$ (Equation (3)) [44]: Its calculation starts from zero at bud-burst (flowering) and stops when an appropriate critical amount of $GDD$ is reached (370 $GDD$ and 50 $GDD$, respectively). $GDD$s are calculated through mean daily air temperature and a base temperature (set to 10 °C for cv. Nebbiolo):

$$GDD_n = \sum_n (T_{av}(n) - T_{base}(n)) \qquad (3)$$

with the assumption that, when the mean daily temperature $T_{av}(n)$ is lower than $T_{base}$, $GDD_n$ is set equal to 0 for that day.

The calculation of the beginning of ripening, veraison, and harvest occurs by means of amounts of $GDD$s (Equation (3)) and critical thresholds of berry sugar content. These thresholds were set in the calibration to 10 °Bx for the beginning of ripening, 12.5 °Bx for veraison, and 25 °Bx for the harvest, which are specific to cv. Nebbiolo.

The leaf area index ($LAI$, m$^2$ m$^{-2}$) is calculated as a measure of plant development [45,46],

$$LAI = \Delta_I \times F_T \times DENS \times I_W \qquad (4)$$

using some functions and coefficients detailed in Supplementary Materials, Section S2 (Equations (S3)–(S5)). In more detail, leaf expansion is simulated by IVINE in terms of *LAI* modulating its value according to the phenological phases: The simulation of leaf development starts at bud-burst and stops at veraison and, from October 1st, leaf senescence is considered by IVINE by imposing a decreasing linear trend. Additional corrections are performed by taking into account the eventual vine trimming carried out in the vineyard.

The value of berry sugar content *BSC* (°Bx), considered a good indicator of maturity and quality, is evaluated by

$$BSC = \sigma_{Brix} BSC_{max} \tag{5}$$

in which $BSC_{max}$ is the maximum berry sugar content (e.g., its value at the harvest), imposed equal to 25.5 °Bx for cv. Nebbiolo. The function $\sigma_{Brix}$ is a normalized number, lower than 1, which is parameterized by means of a double sigmoid curve, a function of thermal time and of cultivar sugar content value at harvest [47,48], whose calculation starts from the phenological stage of flowering (see Supplementary Materials, Section S3).

The yield (kg vine$^{-1}$) is simulated by means of a photosynthetic process, starting from the flowering stage with the following equation [22],

$$Yield = 5.5 \frac{DM_{cluster}}{D\_P} \tag{6}$$

where $DM_{cluster}$ is the dry matter accumulation into vine clusters (see Supplementary Materials, Section S4), $D\_P$ the plant density, and 5.5 an empirical coefficient [22].

Other IVINE outputs are listed in Section S5.

### 2.2. Input Data

To feed IVINE, hourly data of atmospheric and soil variables are required (Section 2.1). Since sufficiently long series of meteorological or agrometeorological data to perform climatological analyses in that zone do not exist, external climatic databases of meteorological observations reconstructed by models and/or measurements were considered. All data but soil variables were directly extracted from the archive of the gridded database GLDAS2.0 (Global Land Data Assimilation System version 2.0 [49]). GLDAS is a global archive created by NASA (Goddard Earth Sciences Data and Information Services Center [50]), whose purpose is to assemble data observed from satellites and ground-based and surface models. Its version 2.0 (GLDAS2.0) contains data from 1948 to 2010 with a spatial resolution of 0.25° in longitude and latitude (about 25 km in the Piedmont region) and a temporal resolution of three hours. GLDAS2.0 data were then interpolated at an hourly rate.

Despite the GLDAS2.0 database also containing soil parameters produced by simulations performed using a land surface model (the NOAH model), we did not use such values. We ran instead the land surface model University of Torino model of land Process Interaction with Atmosphere (UTOPIA) [51], driven by atmospheric GLDAS2.0 data, to recalculate soil variables. The reason for this choice was derived by the results of an analysis [52,53] in which we demonstrated that UTOPIA soil variables (soil temperature and soil moisture) were proven to be closer than GLDAS2.0 ones to the observations carried out during a 3-year experimental campaign [54] carried out in the same area examined in this paper, in particular concerning the highest and lowest values of soil temperature and moisture. The GLDAS2.0 and UTOPIA hourly data used as inputs for IVINE covered a period of 60 years, from 1950 to 2009. Their domain was represented by an area of 15 grid points that included the Langhe and Monferrato wine regions (Figure 1 and Table 1) of Piedmont, characterized by different elevations varying from 95 to 623 m a.s.l. and two different soil textures (loam and clay loam).

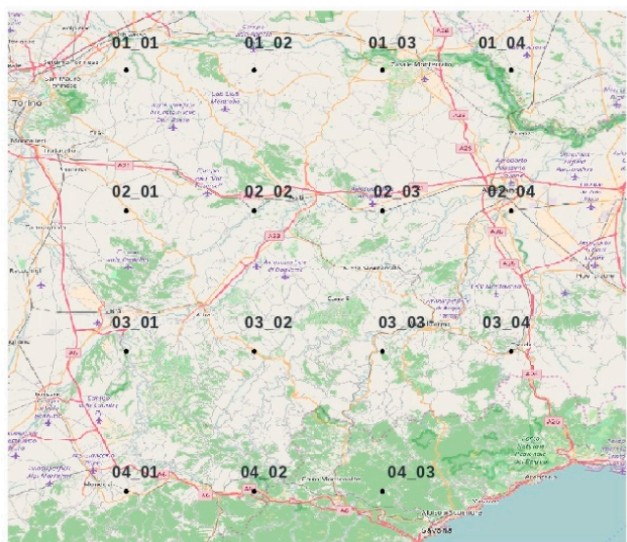

**Figure 1.** Location of the 15 grid points over the Piedmontese territory.

**Table 1.** Coordinates of grid points (longitude and latitude, in degrees and decimals, °E and °N, respectively, elevation in m a.s.l.) and soil texture of grid points considered in the study (the code refers to the [36] classification, also used by the US Department of Agriculture).

| Grid Points | Coordinates | Elevation (m a.s.l.) | Soil Texture |
|---|---|---|---|
| 01_01 | 7.875, 45.125 | 269 | Clay loam-8 |
| 01_02 | 8.125, 45.125 | 207 | Clay loam-8 |
| 01_03 | 8.375, 45.125 | 154 | Clay loam-8 |
| 01_04 | 8.625, 45.125 | 95 | Clay loam-8 |
| 02_01 | 7.875, 44.875 | 257 | Loam-5 |
| 02_02 | 8.125, 44.875 | 181 | Loam-5 |
| 02_03 | 8.375, 44.875 | 153 | Clay loam-8 |
| 02_04 | 8.625, 44.875 | 107 | Clay loam-8 |
| 03_01 | 7.875, 44.625 | 294 | Loam-5 |
| 03_02 | 8.125, 44.625 | 416 | Loam-5 |
| 03_03 | 8.375, 44.625 | 322 | Loam-5 |
| 03_04 | 8.625, 44.625 | 342 | Loam-5 |
| 04_01 | 7.875, 44.375 | 605 | Loam-5 |
| 04_02 | 8.125, 44.375 | 623 | Loam-5 |
| 04_03 | 8.375, 44.375 | 402 | Loam-5 |

*2.3. Model Validation*

A comparison between IVINE outputs and measurements collected during some field experiments carried out in some Piedmontese vineyards was performed. The variables measured were the timing of some phenological stages, the vine *LAI*, the berry weight, and the sugar content.

Measurements were carried out from 2004 to 2010 on the cv. Nebbiolo in three different experimental vineyards located within the most famous wine regions in Piedmont (Langhe, Roero, and Monferrato): Castiglione Falletto (44°37′ N; 7°59′ E; 275 m a.s.l.), Fubine (44°58′ N; 8°26′ E; 200 m a.s.l.), and Castagnito (44°45′ N; 8°01′ E, 300 m a.s.l.). For the first two sites, input data required by IVINE were collected from sensors installed within the vineyards and from regional meteorological stations of Quargnento and Serralunga d'Alba. For the Castagnito site, input data were collected from the regional meteorological station of Castellinaldo and from the global archive GLDAS2.0.

Regarding phenological phases, observations performed in the experimental sites reported the BBCH stage (BBCH means Bundesanstalt, Bundessortenamt and CHemical industry; it is the German scale used to identify the phenological development stages of a plant) achieved at the date of the survey, based on the complete list of BBCH stages [55]. Surveys were performed during the

2008–2010 vegetative seasons at Castiglione Falletto, and during the 2008–2009 seasons at Fubine. IVINE simulations instead returned the dates (in Julian days) in which some BBCH stages occurred (to be precise, the beginning of bud break, BBCH 7; flowering, BBCH 65; fruit set, BBCH 71; veraison, BBCH 83 and °Bx $\geq$ 12.5; harvest, BBCH 89 and °Bx $\geq$ 25; Reference [55]). Despite the attempt to make the surveys in proximity to the beginning of the IVINE calculated stages, sometimes the achieved stage was not in the list of those evaluated by IVINE, making a direct comparison difficult.

Regarding the seasonal evolution of the leaf area index (*LAI*: $m^2_{leaf\ area}\ m^{-2}_{soil\ area}$), available measurements performed every 15–20 days refer to the period May–October 2009 in Castiglione Falletto and Fubine. *LAI* was estimated by comparing the radiation above the top of the vegetation to the one intercepted by the canopy using a solarimeter bar placed within the canopy, selecting the minimum value of radiation of the bar and comparing it to the radiation above the vegetation (see more details in Reference [56]). The vines were generally about 0.5 m thick, and measurements were carried out mostly during the central hours of the day.

Berry weight and sugar content measurements were measured at Castagnito approximately every 10 days in the periods July–harvest of 2004–2005 and July–September of 2006–2007. For each measurement, 200 berries were collected and weighed, and the juice obtained by their pressing was analyzed to determine their sugar concentration (°Bx).

### 2.4. Sensitivity Analysis

To understand the importance of the input data and their role in determining the IVINE output values, a sensitivity analysis test was carried out on main input variables.

Among input data, air temperature and soil water potential were chosen as the primary parameters to be investigated, in order to assess their relevance to model behavior. The impact of input variability was evaluated on the following output variables: Phenological phases, berry sugar content, leaf development, and yield.

One year of input data was the period selected for carrying out this kind of analysis: Since the choice of period and site were meaningless, we arbitrarily chose the last year of the selected time period (1950–2009) and one specific grid point (the one labeled as 03_01, whose details are listed in Table 1). The reason for choosing 2009 was to have the simulation output data in the same temporal period in which the IVINE model was calibrated for cv. Nebbiolo, while the reason for selecting the 03_01 grid point was that it was located at an intermediate elevation and its soil type (loam) was more common in the area.

The values of input temperature were varied in nine scenarios of simulation by summing to all air input temperatures a fixed value $\Delta T_{air}$ respectively equal to −2.0, −1.5, −1.0, −0.5, 0.0, +0.5, +1.0, +1.5, +2.0, where the value $\Delta T_{air} = 0\ °C$ corresponds to no change in input temperature (control run). The values of soil water potential were varied in seven scenarios of simulation by summing to all input values a fixed value $\Delta \Psi$ respectively equal to −3.0, −2.0, −1.0, 0.0, 1.0, 2.0, 3.0 m (note that 1 m of hydraulic head roughly corresponds to 0.01 MPa of suction), where the value $\Delta \Psi = 0$ m corresponds to no change in input soil water potential (control run). Since this variable has two realistic limiting thresholds, e.g., the wilting point and the field capacity, at each step it was checked that the modified values stayed between those thresholds.

### 2.5. Long-Term Simulations and Statistical Analysis

The time trend of each variable and dependence of phenological phases and physiological variables on elevation and soil type were examined for each grid point. Then, results of simulations performed in grid points with different values of soil type and elevation were analyzed and compared, in order to highlight the effects of such variables.

As a general premise in evaluating the results, it is necessary to consider, in the following analysis, that the IVINE model was calibrated for cv. Nebbiolo through comparisons with data recorded in the last 10 years, and thus this calibration (see details in Section S6) is representative of the standard

practices currently performed. When current values are compared to those referring to the beginning of the simulation, the latter should be interpreted as the values of a plant raised similarly to the current plants, but 60 years before. Thus, the output variations can be considered to be the consequence of changes in the input data, e.g., they can highlight more efficiently the effects of climate change. For the same reason, this approach could not take into account the evolution of the change of vineyard methodologies in the analyzed time, in which vine grower standards have certainly changed.

A statistical test related to the significance of linear regression slopes over the whole period was performed on all IVINE output variables. In the paper, only those for the phenological phases, berry sugar content, maximum annual value of leaf area index, and yield are shown. In all cases, the selected test was the Cox–Stuart test, and the significance level was chosen at 95% (*p*-values $\leq 0.05$).

## 3. Results

### *3.1. Model Validation*

#### 3.1.1. Phenological Stages

Since the in-field visits were not continuous, and IVINE does not simulate all BBCH stages, not always was there a correspondence between simulation and observations. For instance, during the first visit at Castiglione Falletto in 2008 (109th Julian day, e.g., April 18th), the achieved stage was the BBCH 11, while the last stage simulated by the model at that date was the BBCH 7 (on April 5th). Thus, the difference in the simulation of the BBCH 7 stage was certainly lower than 15 days. Considering these unavoidable discrepancies, looking at Tables 2 and 3 the typical error of IVINE in predicting the occurrence of phenological stages could be considered in the interval of 5–10 days (underestimation) in Castiglione Falletto (Table 2), and 0–5 days (overestimation) in Fubine (Table 3).

**Table 2.** Comparison between occurrence of simulated phenological stages (with their associated BBCH stages) and BBCH achieved stages in the Castiglione Falletto site.

| Phenological Stage, Castiglione Falletto | Year | Simulated | | Achieved | |
|---|---|---|---|---|---|
| | | Julian Day | BBCH Stage | Julian Day | BBCH Stage |
| Bud-break | 2008 | 96 | 7 | 109 | 11 |
| Flowering | 2008 | 165 | 65 | 161 | 63 |
| Veraison | 2008 | 236 | 83 | 223 | 81 |
| Harvest | 2008 | 300 | 89 | 289 | 89 |
| Bud-break | 2009 | 101 | 7 | 112 | 13 |
| Flowering | 2009 | 148 | 65 | 145 | 61 |
| Veraison | 2009 | 220 | 83 | 213 | 81 |
| Harvest | 2009 | 262 | 89 | 279 | 89 |
| Flowering | 2010 | 155 | 65 | 155 | 63 |
| Veraison | 2010 | 218 | 83 | 207 | 81 |
| Harvest | 2010 | 285 | 89 | 286 | 89 |

**Table 3.** Comparison between occurrence of simulated phenological stages (with their associated BBCH stages) and BBCH achieved stage in the Fubine site.

| Phenological Stage, Fubine | Year | Simulated | | Achieved | |
|---|---|---|---|---|---|
| | | Julian Day | BBCH Stage | Julian Day | BBCH Stage |
| Bud-break | 2008 | 117 | 7 | 120 | 17 |
| Flowering | 2008 | 172 | 65 | 148 | 60 |
| Fruit-set | 2008 | 175 | 71 | 171 | 73 |
| Veraison | 2008 | 244 | 83 | 240 | 83 |
| Fruit-set | 2009 | 164 | 71 | 160 | 73–75 |
| Beginning of ripening | 2009 | 230 | 81 | 224 | 81–83 |
| Veraison | 2009 | 237 | 83 | 224 | 81–83 |

### 3.1.2. Leaf Area Index (LAI)

In the simulations, available data about vine trimming (date and amount of trimming) were given to IVINE and were evident in the results. Comparisons show the quite good performances in Castiglione Falletto (Figure 2), with an overestimation after JD 240 (e.g., the beginning of September). On the contrary, in Fubine (not shown), IVINE underestimated the *LAI* by about 1 m$^2$ m$^{-2}$ during spring (from March to June), but the growth trend was similar to the observed one: In the second part of the season, there was an overestimation (after JD 240) similar to that of Castiglione Falletto. In both cases, IVINE was able (also at Fubine, even if with an initial delay) to simulate well the potential growth of the leaf surface (until it was artificially reduced), while it had difficulties in capturing the slow decrease of *LAI* in the later part of the season.

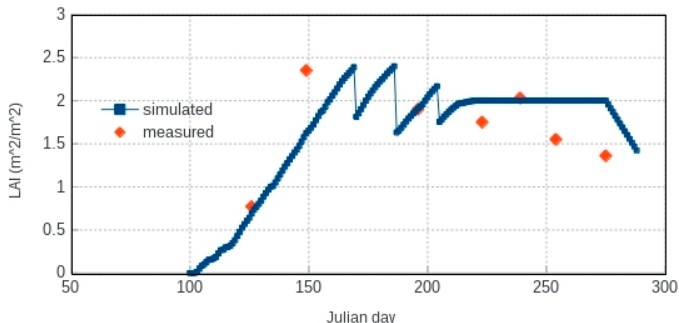

**Figure 2.** Comparison between simulated and measured leaf area index (*LAI*) in the Castiglione Falletto site.

### 3.1.3. Berry Growth

The IVINE model was able to simulate the evolution of berry growth during all examined seasons (Figure 3). The simulated values were generally well reproduced in the first part of the season, with an overestimation in July 2007 (Figure 3b), and were generally underestimated starting from about mid-August, with departures variable in the three years: Small in 2004, 2006, and 2007 (0.1–0.2 g), and larger in 2005 (0.3–0.4 g, Figure 3a). In all simulations, a "jump" of 0.1–0.2 g was present in JD 220: This was the effect of the cluster thinning that, in the absence of recorded information, was imposed on JD 220 of each year, with an intensity of 1 cluster/vine.

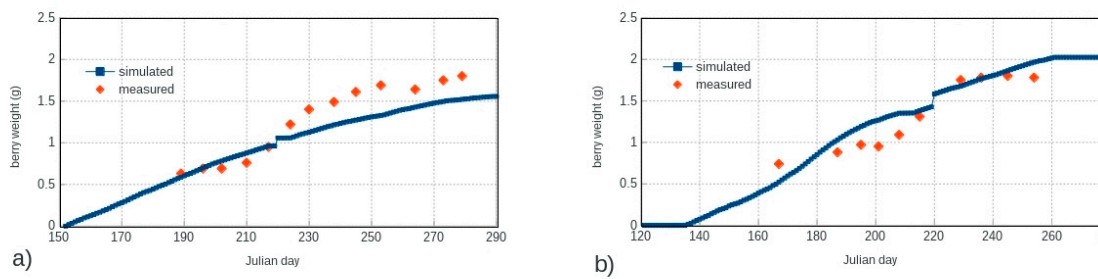

**Figure 3.** Comparison between simulated and measured berry weight (g) in the Castagnito site during 2005 (**a**), and in the Castagnito site during 2007 (**b**).

### 3.1.4. Berry Sugar Content

Regarding the berry sugar content, its trend simulated by IVINE was well reproduced during the whole season and in all years (Figure 4), with minor overestimations (always lower or equal to 2 °Bx) observed mainly between mid-July and mid-August. The simulated sugar content resulted close to the observed values in the central and final part of all seasons, while it was overestimated during the earlier part.

To quantify the above-mentioned intercomparisons, mean absolute error (MAE) between simulations and observations was calculated and averaged every year for *LAI*, berry weight, and berry sugar content (standard deviations refer to the annual average). The values are listed in Table 4.

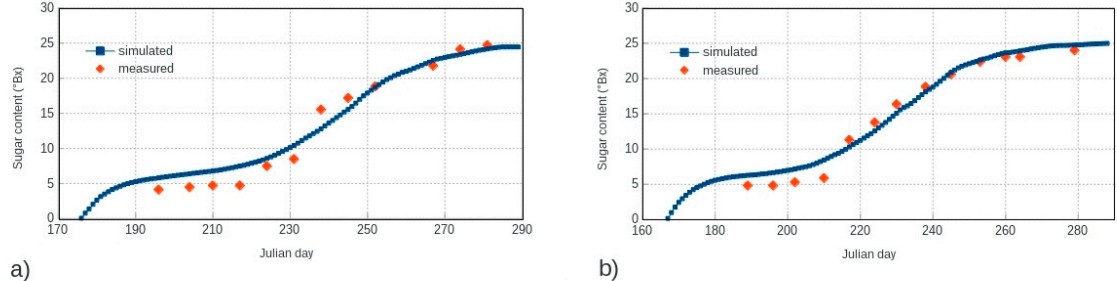

**Figure 4.** Comparison between simulated and measured berry sugar content (°Bx) in the Castagnito site during 2004 (**a**) and 2005 (**b**).

**Table 4.** Comparison between simulated and measured values of *LAI*, berry weight, and sugar content at Fubine. MAE: Mean absolute error.

| Year | Experimental Site | Variable | Average MAE | Standard Deviation |
|------|------------------|----------|-------------|--------------------|
| 2004 | Castagnito | Berry weight (g) | 0.15 | 0.11 |
| 2005 | Castagnito | Berry weight (g) | 0.19 | 0.12 |
| 2006 | Castagnito | Berry weight (g) | 0.16 | 0.08 |
| 2007 | Castagnito | Berry weight (g) | 0.16 | 0.1 |
| 2004 | Castagnito | Sugar content (°Bx) | 1.5 | 0.85 |
| 2005 | Castagnito | Sugar content (°Bx) | 1.12 | 0.7 |
| 2006 | Castagnito | Sugar content (°Bx) | 1.99 | 1.15 |
| 2007 | Castagnito | Sugar content (°Bx) | 1.51 | 0.74 |
| 2009 | Castiglione Falletto | Leaf Area Index ($m^2/m^2$) | 0.31 | 0.3 |
| 2009 | Fubine | Leaf Area Index ($m^2/m^2$) | 0.68 | 0.47 |

Based on those intercomparisons, performed in some experimental sites in Piemonte wine regions, we could conclude that IVINE seemed able to well represent the evolution of phenological phases and physiological parameters for cv. Nebbiolo and in that region.

*3.2. Sensitivity Analysis*

The sensitivity analysis was done on air temperature and soil water potential, and in this section the main results are reported.

Figure 5a shows the results of sensitivity analysis on the date of the flowering stage (expressed in Julian days) as a function of the variation in air temperature ($\Delta T_{air}$). The graph clearly shows the effect of increasing temperature. The flowering stage tended to anticipate for higher values of air temperature. The anticipation was about 8 days for 1 °C of air temperature increment and, as expected, varied almost linearly in the range ±2 °C of $\Delta T_{air}$, without signs of thresholds or saturation.

Other phenological phases showed similar behaviors related to the sensitivity analysis, varying almost linearly with $\Delta T_{air}$ and showing negative trends, except for the dormancy break, which occurred later with increasing $\Delta T_{air}$. Since occurrence of all spring phenological stages but dormancy break anticipated, and dormancy break postponed, with increasing air temperature, the overall effect was a shortening of the period in which the vines prepared for the future vegetative season.

The effects of temperature variations were also analyzed for berry sugar content, evaluated on the 287th JD (corresponding to October 13th or 14th) (Figure 5b). The different simulations show that the sugar content increased with increasing air temperatures, but in this case the behavior was not linear. Around $\Delta T_{air} = 0$ °C, the rate of variation of the sugar content was about 1.1 °Bx °C$^{-1}$. As expected

from Equation (5), Equations (S6), and (S7), above $\Delta T_{air} = 1\ ^{\circ}\text{C}$, the value of sugar content stabilized at the quasi-asymptotic value of about 25.5 °Bx.

Other phenological phases showed similar behaviors related to the sensitivity analysis, varying almost linearly with $\Delta T_{air}$ and showing negative trends, except for the dormancy break, which occurred later with increasing $\Delta T_{air}$. Since occurrence of all spring phenological stages but dormancy break anticipated, and dormancy break postponed, with increasing air temperature, the overall effect was a shortening of the period in which the vines prepared for the future vegetative season.

The effects of temperature variations were also analyzed for berry sugar content, evaluated on the 287th JD (corresponding to October 13th or 14th) (Figure 5b). The different simulations show that the sugar content increased with increasing air temperatures, but in this case the behavior was not linear. Around $\Delta T_{air} = 0\ ^{\circ}\text{C}$, the rate of variation of the sugar content was about $1.1\ ^{\circ}\text{Bx}\ ^{\circ}\text{C}^{-1}$. As expected from Equation (5), Equations (S6), and (S7), above $\Delta T_{air} = 1\ ^{\circ}\text{C}$, the value of sugar content stabilized at the quasi-asymptotic value of about 25.5 °Bx.

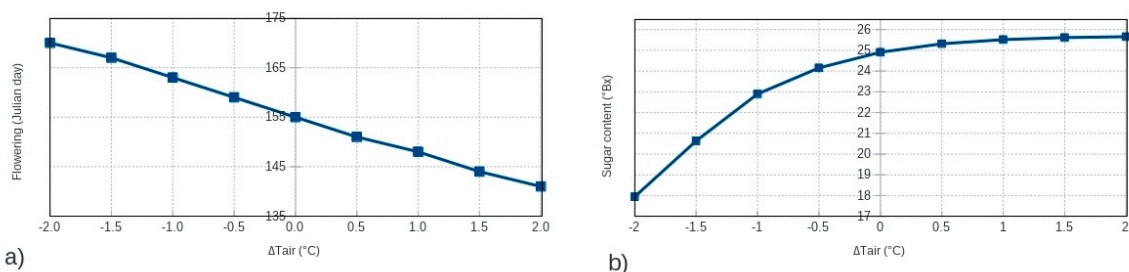

a)   b)

**Figure 5.** Sensitivity to changes of air temperature on the date of the flowering phase (expressed in Julian days) (**a**) and on the berry sugar content (in °Bx) evaluated at the 287th Julian day (corresponding to October 14th) (**b**). $\Delta T_{air}$ is the difference between the input temperature and the actual temperature record.

Figure 6a shows the results of sensitivity analysis related to the soil water potential on the maximum value of the *LAI* reached during the vegetative season as a function of $\Delta\Psi$. The graph shows that the value of *LAI* increased not linearly with increasing $\Delta\Psi$. Given the relation between $\Psi$ and soil moisture ($\Psi = \Psi_s\, q^{-b}$, $\Psi_s$ being the suction for saturated soil, $q$ the soil saturation ratio, and $b$ a coefficient, and $b$ and $\Psi_s$ depending on the soil texture [36]), the abscissae of Figure 6 can also be interpreted as a (nonlinear) soil moisture scale, with the lowest values on the left and the highest values on the right.

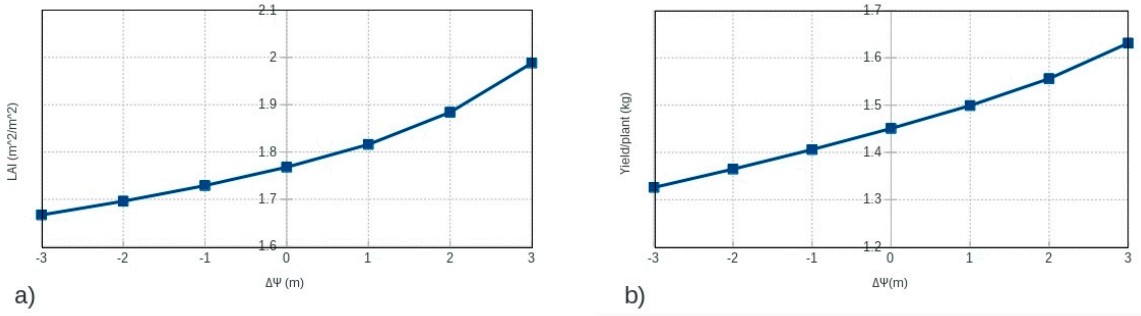

a)   b)

**Figure 6.** Sensitivity to changes in soil water potential on the maximum *LAI* (expressed in $\text{m}^2\ \text{m}^{-2}$) (**a**) and on the yield/vine (expressed in kg) (**b**). $\Delta\Psi$ is the difference between the input soil water potential and the actual soil water potential record. Note that 1 m of water potential (hydraulic head) corresponds to about 0.01 MPa of suction.

With $\Delta\Psi = 0$ m, the variation rate of *LAI* was about $0.04\ \text{m}^2\ \text{m}^{-2}$ for 1 m of soil water potential increment. When $\Delta\Psi > 0$ (<0), the rate was larger (smaller).

The effects of soil moisture variation on the yield were also studied (Figure 6b). The sensitivity analysis highlighted a nonlinear positive trend of the yield with increasing soil moisture. The yield variation was about 0.04 kg vine$^{-1}$ for 1 m of soil water potential, in the range of $\Delta\Psi = 0$ m. As in the case of *LAI*, the rate of growth of the yield vine$^{-1}$ increased (decreased) when $\Delta\Psi > 0$ (<0).

### 3.3. Long-Term Simulations

Due to the impossibility of showing here all results (60 years of simulations carried out on the 15 grid points selected in GLDAS2.0, and 10 relevant variables to show), we decided to comment on figures showing time trends on groups of three grid points with the same soil texture and different elevations, and groups of two grid points at similar elevations and with different soil textures. We did not consider, in our study, the effect of exposition, since the horizontal resolution of the GLDAS2.0 database was too poor to highlight such kind of differences.

### 3.3.1. Effect of Elevation

Generally, the occurrence of all phenological stages showed the same trend: Thus, among all of them, the flowering stage was selected to show the results obtained in this study.

Figure 7a shows the evolution of the flowering date simulated by IVINE in the 60 years (1950–2009) in three sites characterized by elevations differing by approximately 400 m (from the lowest to the highest point). The effect of elevation was evident and seemed to remain constant along the entire analyzed period. Note that the flowering dates at the highest point since 2000 were in the same range as those near 1950 at the intermediate point. This result was in agreement with those relative to the sensitivity experiment on temperature, considering that, usually, temperature decreases about 0.6 °C for every 100 m of elevation.

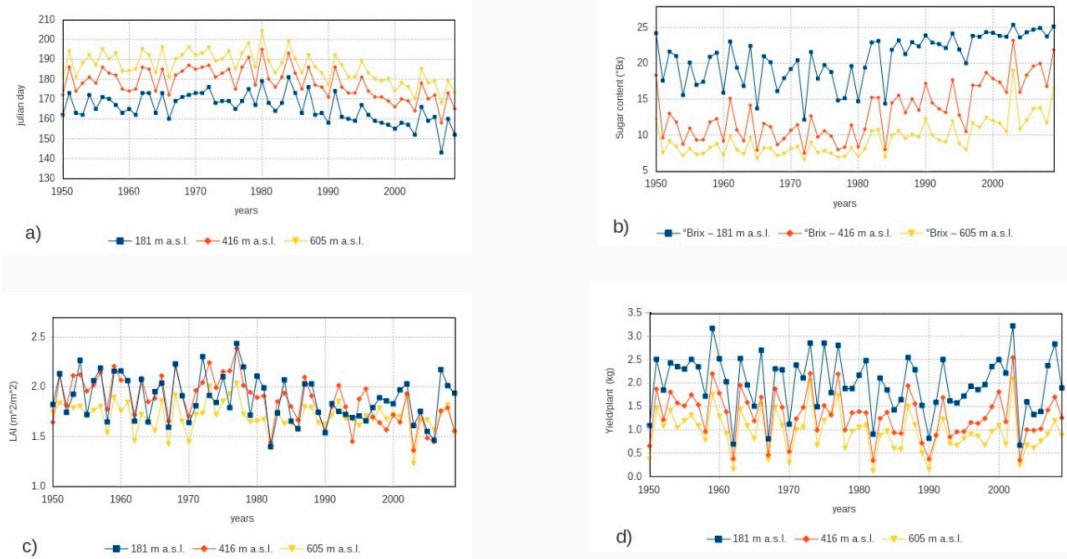

**Figure 7.** Flowering date (expressed in Julian days) (**a**), berry sugar content (**b**) at the date of 287th Julian day (expressed in °Bx) (**c**), *LAI* maximum value (expressed in m$^2$ m$^{-2}$), and yield per vine (expressed in kg) (**d**), simulated by IVINE in three grid points characterized by different elevation. Cv.: Nebbiolo.

Looking at the entire period, the largest variations occurred starting in 1980, which was the year showing the latest flowering date, while the earliest flowering date was observed in 2007. The large anomaly of 2007 was justified by the large positive thermal anomaly during the previous winter and spring over a large portion of Western Europe, more pronounced over northwestern Italy [57]. The trend over the total period evidenced in the simulation was negative, and accounted for about 21 days of anticipation of the flowering stage in the last 30 years.

In Figure 7b, values of berry sugar content simulated at the date of the 287th JD of each year, in the same three sites previously selected, are shown. The effect of elevation was evident also in this case, but, differently from the flowering stage, the difference of sugar content between the lowest elevation site (181 m a.s.l.) and the intermediate one (416 m a.s.l., i.e., 235 m higher) was much larger than the difference between the intermediate elevation site and the highest one (605 m a.s.l., i.e., 189 m higher), especially in the years with the lowest berry sugar content. There were no evident clear trends in the first 30 years in each site, while after 1980 increasing trends were visible, larger for the intermediate elevation site.

The difference in the trends of the two extreme points, evaluated over the whole period, showed a decrease of 2 °Bx for each 100 m of elevation gained. There was also evidence of a significant trend starting from about the 1980s, when interannual variability seemed to decrease in the highest site, with the exception of the year 2003, in which an exceptionally hot and dry summer [58] stimulated IVINE to estimate the highest sugar content of the whole period. Looking at the values at the various altitudes, it is also visible that the vineyards located at the intermediate site had initially very low values of sugar content, comparable to those of the highest elevation site, but starting in 1980 these values increased, almost equaling those of the lowest elevation site at the beginning of the simulations.

Regarding the maximum value of *LAI*, the simulations performed at different elevations are shown in Figure 7c. This variable was related to the vigor of the grapevine, and thus a large value indicated a larger number of leaves per plant, or larger leaves. *LAI* generally increased with warmer temperatures (but could be limited by too hot temperatures, too far from optimal), while it could be limited by too low soil moisture content (e.g., when soil moisture in the root zone approached the wilting point).

At first glance, the effects of elevation appeared less evident than for the previously examined variables. We could notice also for this variable a partition in two subperiods: In the first 30 years, *LAI* maximum values did not vary appreciably, while starting in 1980 there was a decreasing trend at all elevations. In the first period, the lowest and intermediate grid points showed similar values, while starting in 1990 the values of the intermediate grid point appeared more similar to those of the highest grid point. We think that both temperature and soil moisture values, which determine the *LAI* value (Equation (4), Equations (S3)–(S5)), could explain such behaviors, as previously stated. Another evident feature was the decrease of the interannual variability of simulations results since 1980.

The interannual variability of the yield per vine (Figure 7d) was very high, masking any visual trend, but we saw some more stable years in the periods 1953–1958 and 1993–2001. The largest yield at all elevations was observed in 2002, while curiously the lowest yield occurred one year later (2003) at the lowest elevation, and in 1962 at the highest one, and in both years at the intermediate elevation. The effect of elevation was evident among the three grid points, the lowest (highest) one showing the largest (smallest) yield/vine. Differently from the case of the *LAI*, here the grid point at the intermediate level showed yields more similar to those at the highest elevation during the entire analyzed period.

### 3.3.2. Effect of Soil Texture

The following figures show the time trends of the simulations of the same variables previously shown, but referring to two grid points located at very similar elevations but with different soil textures.

Figure 8a shows the Julian days of the flowering stage. Loam soil exhibited slightly anticipated stages with respect to clay loam soil, with differences generally of 1–2 days, which were not significant. Both soils evidenced a clear decreasing trend starting in 1980.

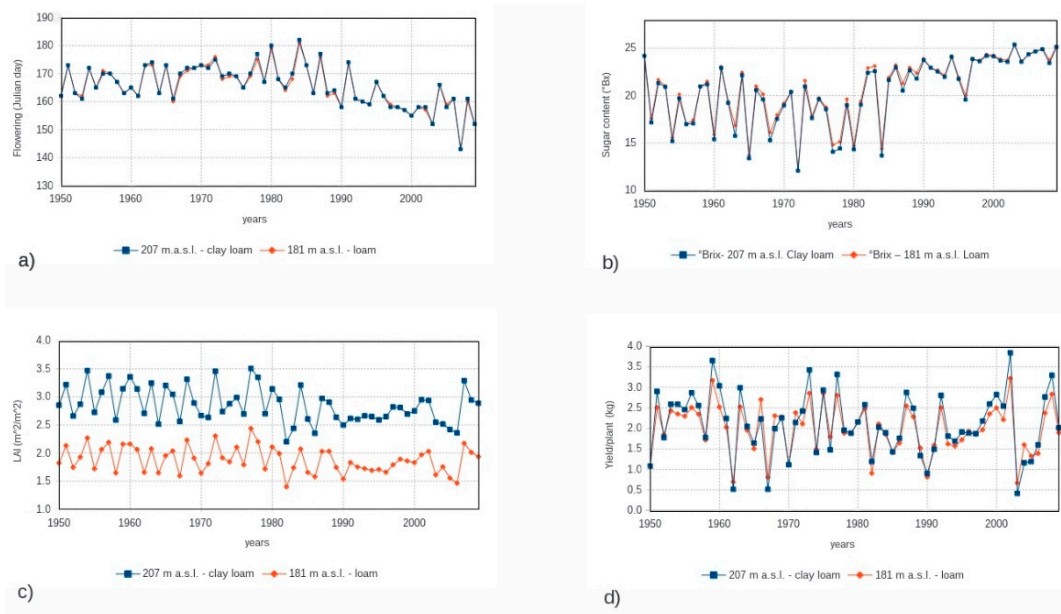

**Figure 8.** Flowering date (expressed in Julian days) (**a**), berry sugar content at the date of 287th Julian day (expressed in °Bx) (**b**), *LAI* maximum value (expressed in $m^2 \, m^{-2}$) (**c**), and yield per vine (expressed in kg) (**d**), simulated by IVINE in two grid points characterized by different soil texture. Cv.: Nebbiolo.

Figure 8b shows the time trend of the berry sugar content, evaluated on the 287th JD, for the two soil textures. Loam-type soil sometimes exhibited higher values of sugar content with respect to clay loam-type, the differences being limited to 0–3 °Bx, larger in the first 35 years of the simulation. Starting in 1984, the interannual variability of the sugar content dropped to its minimum values, and a clear increasing trend was present. During this period, only two years showed values larger than 25 °Bx: 2003 and 2009 (the last year of the simulation). A value of 18 °Bx, which can be associated with the occurrence of the berry softening phenological stage (BBCH 85) in current vineyard management for cv. Nebbiolo, was reached 18 times in the period 1950–1979 and 28 times in the period 1980–2009. The last time at which the 18 °Bx threshold was not reached was the year 1984.

In Figure 8c, the results of the simulations of *LAI* are reported. Differently from the previous figures, here the effect of different soil texture is evident: The grid point characterized by a clay loam-type soil showed the largest values of *LAI* during the whole period, with a systematic shift of about 1 $m^2 \, m^{-2}$ above the values for loam soil. This was mainly caused by the different soil moistures in the two soil types (the soil saturation ratio is larger in loam soil due to its higher hydraulic conductivity). Values starting in 1980 were lower than in the previous period, almost equaling the minima during 1950–1979 (about 2.5 $m^2 \, m^{-2}$ for clay loam soil), and interannual variability was very low during the period 1989–2002, perhaps due to the reduced effect of the combined variation of meteo-climatic parameters on maximum *LAI*.

The yield per vine (Figure 8d) evidenced that the values associated with the clay loam soil type were, on average, higher than those associated with loam soil due to higher values of soil water potential (in absolute value): In fact, since saturated soil water potential was more negative for clay loam soil (code 8), we expected higher soil moisture in such soils, and then higher yields. In this case, the differences were small but discernible (less than 0.5 kg vine$^{-1}$), and seemed larger when yield was larger. As already noted for the *LAI*, the period 1989–2002 was characterized by increasing yields with an extremely small interannual variability, due to the combined variations of meteo-climatic variables.

As a general conclusion for this section, the values belonging to simulations carried out on different soil types at an almost-the-same elevation showed that, compared to the elevation, the soil type played a smaller role. The differences were very low and could be slightly correlated with the

soil moisture content, which was higher for clay loam soil, since saturated soil water potential and porosity are higher (in absolute value) in clay loam soil.

*3.4. Slopes of Regression Trends*

The following tables (Tables 5 and 6) contain the slope of the time trends obtained from the linear regression of the simulations over the whole time period, from 1950 to 2009 (Table 5), and over the most recent 30 years, from 1980 to 2009 (Table 6). As a preliminary note, we say that the time trends of the period 1980–2009, related to almost all output variables considered, resulted as statistically significant (see Section S7) and different with respect to the time trend of the whole time period. This result highlighted a sensitivity of IVINE to its input data starting in 1980, and it could be assumed to be a sign that climate change started to have significant effects on Piedmontese vineyards starting in 1980.

**Table 5.** Linear regression slopes evaluated for the variables discussed in the text over the whole 60-year period, 1950–2009, in three grid points with the same soil texture and different elevations.

| Variable/Elevation | 181 m a.s.l. | 416 m a.s.l. | 605 m a.s.l. |
|---|---|---|---|
| Flowering Stage (JD year$^{-1}$) | −0.2 | −0.2 | −0.2 |
| Berry Sugar Content (°Bx year$^{-1}$) | 0.1 | 0.1 | 0.1 |
| *LAI* Maximum Value (m$^2$ m$^{-2}$ year$^{-1}$) | −0.004 | −0.009 | −0.003 |
| Yield (kg year$^{-1}$) | −0.004 | −0.005 | −0.005 |

**Table 6.** Linear regression slopes evaluated for the variables discussed in the text over the most recent 30-year period, 1980–2009, in three grid points with the same soil texture and different elevations.

| Variable/Elevation | 181 m a.s.l. | 416 m a.s.l. | 605 m a.s.l. |
|---|---|---|---|
| Flowering Stage (JD year$^{-1}$) | −0.7 | −0.7 | −0.7 |
| Berry Sugar Content (°Bx year$^{-1}$) | 0.2 | 0.3 | 0.2 |
| *LAI* Maximum Value (m$^2$ m$^{-2}$ year$^{-1}$) | −0.001 | −0.008 | 0.000 |
| Yield (kg year$^{-1}$) | 0.007 | 0.008 | 0.004 |

The slope coefficients of the flowering phenological stage (as well as those of all other phases, not analyzed in this paper) were negative in both considered periods (1950–2009 and 1980–2009) and for all analyzed grid points. Those related to the most recent period were larger and evidenced a quickly decreasing trend (about three weeks of anticipation in 30 years).

The same consideration was valid for the berry sugar content, but with positive slope. Considering the most recent 30 years, the positive increasing trend corresponded to an increase of about 6–9 °Bx, which turned out to be quite consistent.

Due to the results of our sensitivity analysis for berry sugar content, we were expecting that this trend would slow down or even stop if temperatures continued to increase, since it was assumed to "saturate". However, it became quite unusual to have particularly low values with higher temperatures.

The *LAI*, as already observed in commenting on Figures 7c and 8c, showed a quite large interannual variability that masked the slopes. The largest negative slope was observed at the intermediate elevation grid point. The lowest grid point in recent times showed a very small negative slope. The highest grid point also did not show any slope in the most recent period. Even considering the most negative slope (intermediate elevation, most recent period), the total *LAI* decrease in 30 years was less than 0.25 m$^2$ m$^{-2}$.

Regarding the yield, its variations considering the whole 60-year period or the most recent 30-year period appeared opposite in sign and similar in amplitude. Recent slopes were positive and larger in the lowest and intermediate elevation grid points, with cumulative values of about 200–250 g of increment in 30 years, but this growing rate was limited by the quite low values recorded in the period 2003–2006 (Figure 7d).

The results of the statistical significance of linear regression slopes are listed in Table S1.

The linear regression slopes of flowering, fruit-set, beginning of ripening, and dormancy break stages resulted as significant for all grid points. Concerning the bud-break and veraison stages, most of their linear regression slopes were significant. Regarding the harvest stage, the linear regression slopes resulted as significant only for five grid points.

Regarding physiological outputs, berry sugar content and *LAI* maximum value linear regression slopes were always statistically significant (but in two grid points for *LAI*). Linear regression slopes of yield/vine instead resulted as statistically significant only at three grid points, due to its very large interannual variability.

## 4. Discussion

We validated IVINE by searching observational datasets gathered in field measurements carried out in cv. Nebbiolo vineyards. Despite this vine being widespread in the Piedmont region, there are not so many data suitable for checking IVINE reliability in reproducing the physio-phenological variables mentioned in the paper. Due to the impossibility of finding in the literature similar studies relative to cv. Nebbiolo and performed using other crop models, we compared our results to other recent results obtained for other red wine varieties in zones with climates not too different from the Piedmontese one.

Due to the missing contemporaneity between the experimental phenological stages (observed by two of us during twice-monthly visits) and those calculated by IVINE, there were some difficulties in validating the phenological stage occurrence, since sometimes it was impossible to reconstruct the exact day of some stages. Considering this problem, based on the results presented, we could evaluate the typical error of IVINE in predicting the occurrence of phenological stages in the interval −5–+10 days (see Tables 2 and 3). These values generally agreed with those found in the literature. For instance, Cola et al. (2014) [22], who analyzed the performance of their model on a vineyard of cv. Barbera in Italy, found a mean value of MAE of 0.7 BBCHs and yearly MAEs of 0.6–1.1 BBCHs, roughly corresponding to about 5–10 Julian days, similar to our values. Fraga et al. (2015) [27], who studied Portuguese red grapevines using the STICS model, found for the flowering phase a MAE generally lower than one week, with annual differences up to 13–17 days, and a higher accuracy for the harvest stage, with an overall MAE of a few days and yearly differences ranging from −7 to +2 days. Valdés-Gómez et al. (2009) [26], examining vineyards in France and Chile, found differences ranging from −6 to +1 days for the flowering stage and −4 to +4 days for the harvest stage, with an accuracy slightly superior to our values.

The MAEs for the IVINE simulations of berry weight were in the range 0.15–0.19 g. These values appeared lower than those of Mirás-Avalos et al. (2018) [59], who found for Tempranillo grapevines in Spain an MAE of about 0.47 g of dry mass per berry, and also those of Valdés-Gómez et al. (2009) [26], who found for French and Chilean sites MAE values of 0.1–0.5 g.

Regarding *LAI*, our MAEs were in the range 0.31–0.68 $m^2$ $m^{-2}$, and appeared to be slightly smaller than those obtained by Valdés-Gómez et al. (2009) [26] in their study, where the *LAI* absolute bias was 0.5 $m^2$ $m^{-2}$, with individual deviations larger than 1 $m^2$ $m^{-2}$.

In conclusion, the analysis of the validation experiments showed that IVINE performed at least similarly to other published crop models, even if it was impossible to find simulations for the same variety. Phenological phases seemed to be the variables that were less accurately predicted. In our opinion, the use of functions related only to air temperature, with specific thermal thresholds, at least for the first phases, could introduce some approximations that could cause such disagreements. In further studies, the influence of other variables on phenological phases could be taken into consideration.

This was also the main reason why the air temperature was the most sensible IVINE variable among its inputs, and explained why phenological stages anticipated almost linearly with increasing temperature, differently from other outputs. On the contrary, the quasiasymptotic threshold of about

25.5 °Bx, shown by the berry sugar content during sensitivity to air temperature, was an effect of the double-sigmoid curve (and the related parameter's choice) used in the parameterization of such variables (Equation (5), Equations (S6), and (S7)). Further investigations to be carried out in warmer climates and with other varieties could confirm if such a parameterization, adopted for IVINE, could be valid also in warmer climates.

The simulation of the harvest stage deserves a special consideration. IVINE simulates the harvest stage when the berry sugar content reaches a value of 25 °Bx: This value represents the harvest threshold in the current vineyard management practices for cv. Nebbiolo-making. Simulation results in other periods showed the behavior of plants raised using current practices, but with a past climate. They evidenced that this stage was rarely, or never, reached during the first 40 years of the analyzed period, particularly for grid points characterized by high elevations. This result could also be interpreted in another way. If 40 years ago it was impossible to produce Nebbiolo wine with the actual management practices in most of the Piedmont region, now it has become possible, mostly due to temperature increments connected with climate change.

Regarding the long-term simulations, the relevant decrease of yield (per plant) with elevation could be explained by considering that it was related to vegetation photosynthesis, which depends on radiation, temperature, and soil moisture. Among these effects, in our simulations the temperature was the factor changing most effectively in higher elevations. On the one hand, temperature delayed the flowering phase by more than 20 days, on average, considering our highest and lowest grid point (Figure 7a), thus postponing the growth of the berries. On the other hand, the photosynthetic activity was related indirectly to the temperature, and the difference mattered even if the quote difference between the highest and lowest grid points was only 424 m.

The linear regression slopes of most pheno-physiological variables examined during the long-term simulation showed that, for most of these variables, they were significant for most or all grid points, the only exception being the harvest stage (already discussed above) and yield. For the latter, the large interannual variability in each site was noticeable. The anticipation of the phenological phases looked numerically similar to the values reported by Tomasi et al. [31] for a study on a shorter time period in another northern Italy wine region (Veneto), but with different cultivars.

We considered the possible effects of volcanic eruptions on those data: The three volcanic eruptions with some discernible effect on global mean surface temperature in that period were in 1963 (Mt. Agung), 1982 (El Chichón), and 1991 (Mt. Pinatubo) [60]. These eruptions did not cause, in the analyzed region, temperature variations larger than those associated with natural interannual variability, and thus did not have discernible effects on the vineyard variables examined here.

The IVINE crop model, created to study some physio-phenological processes in vineyards on the basis of some micrometeorological and soil observations, and calibrated for cv. Nebbiolo, was able to give a realistic representation of such processes with quantitative data. IVINE could be applied as an instrument that gives to the winegrower some additional data useful for vineyard management. The results of our long-term simulation also showed that IVINE, once adequately calibrated also for other cultivars, could also be used, as we did, to show the effects of climate change on the variables affecting wine production. The use of regional climate model simulations, instead of past and recent observations, as inputs for IVINE could also allow for seeing the expected effects of future climate change on wine variables.

## 5. Conclusions

The crop growth model IVINE was developed to simulate grapevine phenological and physiological processes related to environmental conditions. It requires a set of meteorological and soil data as boundary conditions. The main model outputs are main phenological phases, leaf development, yield, and sugar concentration. At present, the model has been optimized and validated only for cv. Nebbiolo.

IVINE was validated using the data of phenological phases (bud break, flowering, fruit-set, beginning of ripening, veraison, and harvest) and physiological parameters (*LAI*, berry weight, and berry sugar content) recently observed in some Piedmontese vineyards. The model results were accurate in representing both the time trend and the numerical values of pheno-physiological variables related to the specific type of vine cultivar, with accuracies similar to, and in some cases higher than, those of other recent studies.

A sensitivity analysis was performed on air temperature and soil water potential inputs due to their relevance in the model equations, with other variables having minor effects. With increasing temperatures, phenological stages anticipated almost linearly (about 8 day $^\circ$C$^{-1}$), while the berry sugar content showed a nonlinear increase (about 1.1 $^\circ$Bx $^\circ$C$^{-1}$), tending to stabilize its values at the quasiasymptotic threshold of about 25.5 $^\circ$Bx. With increasing soil water potential, the *LAI* showed a nonlinear incremental rate (about 0.04 m$^2$ m$^{-2}$ m$^{-1}$), and the yield showed a not linear positive trend (about 0.04 kg m$^{-1}$).

Long-term simulations, driven by GLDAS2.0 climatological atmospheric data and UTOPIA land surface model soil variables, were performed, running IVINE over 15 grid points for 60 years (1950–2009) within a selected Piedmontese area prone to viticulture. The results indicated significant trends of almost all variables related to physiology and phenology, combined with (for most variables) a reduction in interannual variability, particularly evident for berry sugar content in recent years. These results seem to indicate a strong influence of climate change, at least since 1980, after which almost all variable trends consistently increased.

The future perspective of this project will be the optimization of the IVINE crop model for other grapevine varieties and the execution of simulations in different regions of Italy or the world under current, past, or future climates.

**Supplementary Materials:** The following are available online at http://www.mdpi.com/2073-4395/9/2/94/s1, Section S1: Equations to Extrapolate Hourly Temperatures, Section S2: Parameterization of LAI, Section S3: Parameterization of Berry Sugar Content, Section S4: Parameterization of Yield, Section S5: Other Outputs of IVINE, Section S6: IVINE Calibration, Section S7: Slopes of Regression Trends of All IVINE Outputs; Table S1: Linear regression slopes of the main IVINE variables, evaluated over the full 60-year period. Phenological stage regression slopes are expressed in JD year$^{-1}$, sugar content in $^\circ$Bx year$^{-1}$, *LAI* maximum value in m$^2$ m$^{-2}$ year$^{-1}$, and yield in kg year$^{-1}$. Bold values represent statistically significant trends. For some phenological phases (veraison and harvest), the stage was not reached during several years, and thus the trend was not evaluated (we have put the *** symbol in such cases).

**Author Contributions:** V.A. and C.C. have contributed in equal measure to the research article. T.L.I. and F.S. have managed data acquisition and checking.

**Funding:** This research was partially funded by "MACSUR—Modelling European Agriculture for Food Security with Climate Change, a FACCE JPI knowledge hub", funded for an Italian partnership by the Italian Ministry of Agricultural Food and Forestry Policies (D.M. 24064/7303/15). The Lagrange Project—CRT Foundation/Isi Foundation and IWAY s.r.l. supported two grants of one year for developing the project.

**Acknowledgments:** We acknowledge Silvia Cavalletto, Silvia Ferrarese, Silvia Guidoni, Massimiliano Manfrin, Elena Mania, and the team of IWAY s.r.l. for useful discussions about agronomic and instrumental details, and Hualan Rui for help in managing the GLDAS dataset. We also acknowledge the scientific team that made available the GLDAS dataset in both versions for research purposes.

**Conflicts of Interest:** The authors declare no conflicts of interest.

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
