# Peer review of "Description and Preliminary Simulations with the Italian Vineyard Integrated Numerical Model for Estimating Physiological Values (IVINE)"

_agronomy, doi:10.3390/agronomy9020094_

Round 1
Reviewer 1 Report
The revised version of the manuscript entitled “Description and preliminary simulations with the Italian Vineyard Integrated Numerical Model for Estimating Physiological Values (IVINE)” by V. Andreoli, C. Cassardo, T. La Iacona, and F. Spanna (Ref. agronomy-423553) represents a great improvement from the original version. Authors performed a huge work in revising the text and including new information, which was extremely necessary for the correct understanding of the undertaken work. I thank you for this effort.
One of my main concerns on the original version was the lack of a validation of the model outputs with field measurements. In this revised manuscript, authors included this information.
Nevertheless, English needs great improvements all over the manuscript for the right comprehension of the manuscript, not only about sentence structure but also about wording since some terms used by the authors are not correct, at least from an agronomical point of view (being Agronomy the target journal, this should be corrected).
Apart from these encouraging comments, I feel that the manuscript should be rejected since most of my questions and remarks have not been successfully addressed by authors. I do not know why but they insist on the fact that they developed a new model, when they took equations and even parameters from other models already in use. To me, this is a clear weak point of the manuscript, which is extremely long, by the way.
Therefore, I still suggest the rejection of this manuscript since it does not reach the high-quality standards to be published in Agronomy.
I spent a lot of my time reviewing both the original and this version of the manuscript and have so much comments that I try to summarize in the following pages.
Specific comments (The number of lines “L” refers to the revised version with no track changes, namely, the clean version):
Abstract:
L18-21: “IVINE model simulates physiological and phenological vineyard processes with physically based parameterization of most important processes taking place into vineyards, allowing to understand plant conditions at the microscale”, this sentence is too long and confusing.
L21: “at the microscale”, in your answer you indicate that you refer to the vineyard as “microscale”. I think it is better to say “at the vineyard scale” or “at the grapevine level”.
L23-24: “to set some experimental parameters”, which are?
L25: “a variety living mostly” must be substituted for “a variety grown mostly”.
L29: “soil moisture potential”, this term is not correct, it should be “soil water potential”.
L30: “phenological phases decrease”, maybe you refer to the duration of the phenological stages and not to these phases themselves.
Introduction:
Again, this section is just a compilation of information and citations referred to crop models, without a clear focus on the subject of the manuscript and the work to be undertaken and presented in it. Several of my main concerns regarding this section have not been addressed. For instance: What are the drawbacks of the existing models? How can they be solved? What will your work add to the current body of knowledge?
You make reference to the STICS model (L65-71) that has been adapted to grapevine (L80-83); however, you do not mention that your model (IVINE), which is as new as you claim, is a further adaptation of STICS because it uses several of the modules of this generic crop-model. So, what is the novelty of your work?
Again, English must be corrected since it can mislead readers. For instance, L78-79: “The software STELLA was implemented to predict the daily carbon balance and the dry matter accumulation in grapevine [23]”. This is clearly incorrect, it should read: “A model for predicting the daily carbon balance and the dry matter accumulation in grapevines has been implemented using the STELLA environment [23]”, which is more close to reality. Besides, what does it matter in which environment is the model implemented? You never indicate what programming environment you used for IVINE.
L72-79: This paragraph is a clear example of my main concerns. Here, authors cite several models for grapevine but they do not provide reasons for not using them and instead developing a “new” one (IVINE).
In L84-92, authors talk about modelling phenological stages in grapevine; however, they do not conclude which model is better or why they must develop a new one.
The paragraph from L93 to L99 is an example of how authors talk about something (in this case, climate change influence on grapevine phenology) but they do not finish with a clear point about this subject or how modelling would help predict or counteract the effects of climate change on grapevine phenology.
Materials and Methods:
The authors added further information but they did not change some English mistakes that would impede the proper understanding of their methodology.
For instance, authors answered me that the empirical equations for calculating soil hydraulic properties from variables such as organic matter (not just “organic” as appears in L127 of the revised version of the manuscript), sand and clay percentages are the most accurate method for obtaining those hydraulic variables. However, pedotransfer functions are very local, as proven by many studies and the authors are using here equations developed for the soils of the United States and not for Italian soils. I am not satisfied with their response.
L130: “soil layers number, texture, and depth”; these parameters do not depend on the cultivar but on the soil.
L134: Remove “grapevine”.
L135: “cultivar” instead of “grapevine”.
L144: Why starting in August 1st? The references you cite are for apple trees, would it be the same date for grapevines? Moreover, in your response to this comment, which I made on my previous report, that you obtained GDH by the method of Richardson and provided two references. Why does this answer not appear in the text of the revised version of the manuscript?
L146: Why Q is set to 2.17?
L148: What is the value of this “critical sum”?
L154: “370 GDD and 50 GDD”, do you restart the count of GDD? I mean, after 370 GDD do you count 50 GDD from zero or you stop at 420 GDD?
L159-161: You must indicate how these values were obtained.
L163: How is soil water potential calculated?
L168: Here, you cite a value of a parameter obtained directly from the original version of the STICS model. How was this parameter adapted to grapevine?
L170-172: This should be further explained.
L179-184: How were these equations adapted to Nebbiolo cultivar?
L191: “by means of a curve”, how was this curve constructed?
L193-196: Messy, it must be re-phrased.
L199: Where does this misterious “empirical equation” come from?
Figure 1 must be improved. In some cases numbers are very close to the points and to each other. Since you provide a reference in table 1 (column named “grid points”), maybe you could use the same reference in the figure instead of giving the altitude and soil texture.
L255-286: English must be improved. How many data for each variable do you have, actually? How were field data compared to simulations?
L292: Use “soil water potential”, “soil moisture potential” is not correct.
Results:
L330-331: This can be removed.
L343-344: The underestimation of phenological stages seems rather important. However, no statistical comparison has been performed.
Figure 2 must be improved, it seems like an screenshot. Besides, no statistical assessment of the goodness-of-fit has been performed.
Figures 3 and 4 proved that berry growth is not correctly simulated by the model, with deviations of 14% around harvest date, both under or overestimating measured values.
L394-397: Check English and re-phrase.
L401-403: I agree with you about the statement that the model reproduces well the evolution of phenological stages and berry growth and sugar content over the growing season. However, to me, the errors shown in table 4 are rather high.
L406: “soil moisture potential” is not correct.
L430-431: Remove this sentence.
L433-438: This is confusing.
L447: I still do not understand how soil water potential is measured in meters. Meters of what? Since this is a pressure or a tension, usually the units are bars or pascals (Pa, kPa, MPa) but not meters. I wonder also that, being WATER an international journal, you should use the units from international system of units. Please, check this matter.
L457-552: My remarks that I made in the former version of the manuscript concerning this section have seldomly been accounted for.
Discussion:
L609: “IVINE is one of the few models specifically developed for vineyards”, well I do not agree since most of the equations in this model come from general crop models such as STICS. The other equations come from models already designed for grapevines (for instance, those of the phenological stages) that were calibrated for Nebbiolo cultivar. Therefore, the model is not so novel as claimed by the authors and is not exactly developed for vineyards, but some modules are adapted to vineyards.
L610-611: “physically based equations”, untrue since most of them are empirical.
L614: What do you mean by “more diffuse”?
L615: “we decided to develop a new crop model”, but this is not exactly true since most of the equations used come from other models.
L617-622: This paragraph must be re-written after a careful check of the English language.
L623-639: Although English can be improved and the paragraph can be reduced, this is interesting to see that your model did not perform better than others.
L641-642: How? In the article by Mirás-Avalos et al. (2018) there are no MAE values for berry.
L657: The double-sigmoid curve is used for berry growth and sugar content accumulation in all grapevine varieties. The parameters you used for characterizing Nebbiolo variety in this curve are the reason for the threshold of 25.5 ºBx. In warmer climates would occur the same if you use the same parameters for the curve, the threshold of 25.5 ºBx would never be surpassed. The point is when this threshold is reached. In warmer climates, the threshold should be achieved earlier than in cooler climates.
L660-668: This discussion is rather strange because authors imply that Nebbiolo wine could not be produced in this region 40 years ago.
L669-676: This paragraph seems out of the main line of the discussion.
Conclusions:
This section is too long. Please, reduce it.
References:
Check for mistakes, I have detected some.
L827: “conception” instead of “concption”.
L844: “vitis vinifera” should be written as “Vitis vinifera”.
Author Response
We have attached a file containing a point-by-point response to the reviewer’s comments.

Reviewer 2 Report
This manuscript presents a comprehensive grapevine model that encompasses the calculation of phenology stage, soil water content, leaf area development, berry weight, sugar content, and yield. This approach allows to study the effects of temperature and soil water content on grapevine development and yield and berry composition.
General comments:
1) I believe the objectives of the paper should be made clearer in the introduction. Why do you want to develop a new grapevine model and what are the questions that you would like to answer with this model rather than a modelling exercise?
2) Overall the current paper is a bit lengthy and discursive. The authors may would like to improve the clarity of the paper and make it more concise. For example, in lines 192 to 196: The water limiting factor WL is obtained by comparing the soil water content with two thresholds depending on field capacity and wilting point, normalized by the saturated soil potential; it is equal to 1 if the soil water content is between the two thresholds, while it varies linearly from 0 to 1 or vice versa between the thresholds and field capacity or wilting point, and it is null below the lowest threshold and above the highest threshold. This description is really very hard to follow. The authors can just add an equation in the text and say water limiting factor WL varies linearly from 0 to 1 between wilting point and field capacity.
3) The authors should improve the model description part to make it clearer.
a) Use clearer undernote, like Tmax for maximum temperature, Tmin for minimum temperature. In equation 4, do not use the same note T(h,n) at both sides of the equation, you can change one into Tair or something like this.
b) In equation 3, why do the authors calculate T(h,n) like this if the authors use hourly climate data as input? The authors can directly use hourly temperature and input into equation 4. If you use equation 3, then use daily data as input is enough.
c) I am a bit hesitate about Equation 6. Is this a good way to calculate predawn leaf water potential? some authors think predawn leaf water potentials should equivalent with the most humid layer (Deloire A, Heyns D. 2011) while others found it may not equilibrate with soil water potential. I feel the authors may would like leave their claim that this model can simulate predawn leaf water potential, first of all their no leaf water potential presented in this paper, second the model is not designed for simulating leaf water potential.
Donovan L, Linton M, Richards J. 2001. Predawn plant water potential does not necessarily equilibrate with soil water potential under well-watered conditions. Oecologia 129, 328-335 doi: 10.1007/s004420100738.
Deloire A, Heyns D. 2011. The leaf water potentials: Principles, method and thresholds. Wynboer 265, 119-121.v
d) Eq. 7 How does water status affect leaf area development? The authors should provide a specific equation for this. If the authors feel there is too many equations and figures, some of them can move to supplementary data if they do not hit understanding.
e) Eq 10 give clearer explanation that which parameter correspond to the cultivar sugar content value at harvest.
f) Eq 11 add equation about how PNA, TL and WL are calculated. Do not let readers to guess. This also leads to wonder the result in Fig. 17 and 18, where big differences in leaf area between the two sites but there were no difference in yield. I would like to find the causes in equation but the equation was too vague. For the unit of water potential, I believe the international unit is MPa or kpa, seeing the water potential expressed in meter is less common now. The authors may would like to change their unit for water potential.
g) Eq 12, about the range of Csink, should define in an equation more explicitly. Further in line 200, ‘its range is 0 1 ’ looks strange, why not just say ‘its range is 0 to 1’. I also feel the value of Csink is a bit too simplified. A normal growth vine, even there is no sink limitation, would still have carbon reserve build up in trunk, root, and shoot during the berry ripening period. The authors may check the carbon reserve dynamic literature on grapevine, and give a more reasonable allocation value.
h) Eq 14 just would like to comment on the value of K. it is true that K is influenced by the fraction of intercepted radiation. However this is partly true as for row crops with high canopy like grapevine, wind also plays a big role on transpiration. It is probably very complicated to correct for the wind effects, but could be a point to work on in future.
i) Eq 15 the authors give a complicated fit for berry water content based on GDD, but water content is very close to 1 minus sugar content which was calculated in Eq 10. The authors could use the simple form 1 minus sugar content to calculate the water content first and compare with the data, and add a correction factor if needed.
4) The author may would like to improve the figures a little bit. Here are few suggestions: 1. Add the mean absolute error in table 4 to the corresponding figure and remove table 4; 2. Remove the title of the figure. 3. Make the border of the figure into a solid black line. 4. Combine the figures into one figure with different panels for the same topic, like combine fig. 3 and fig. 4, fig. 5 and fig. 6, fig. 7 and 8, fig. 9 and 10, fig. 11 and 12 etc.
Specific comments:
a) Line 18, ‘being vines generally strongly sensitive to meteorological conditions.’ Could be removed.
b) line 19 I did not think most parameters are ‘physically’ based. The authors may would like to reduce their tone on this or remove this claim.
c) line 20, ‘ It requires a set of meteorological data as boundary conditions.’ Change into what are the model input: meteorology data and soil water status
d) line 22, remove ‘predawn leaf water potential’, this is not the main model output.
e) line 23, the phrase ‘experimental parameters’ is a bit unusual. Maybe just say variety information or parameters.
f) Line 24 ‘Nebbiolo grapevine’ is not a common expression, change into Vitis vinifera L. Nebbiolo for the first time and cv. Nebbiolo afterwards throughout the paper.
g) line 29, ‘soil moisture potential’ is not a common phrase, change into ‘soil water potential’ throughout.
h) L39 the first sentence is quite a general comment which could be removed.
i) L43 weather climate, is duplicated remove weather.
j) L52 add the author name before citation 10.
k) L80 STICS grapevine has appeared for about 10 years, remove ‘recently’. The authors could add a cite to a most recent grapevine model based on bio-physiological processes: Zhu J, Génard M, Poni Set al. 2018. Modelling grape growth in relation to whole-plant carbon and water fluxes. Journal of Experimental Botany, ery367-ery367 doi: 10.1093/jxb/ery367
l) L105 I think the most important variables that tested are air temperature and soil moisture.
m) L114 ‘it works by means of a daily time step’ change into it runs on daily step
n) L119 to L122, simplify this expression, and move it after explaining the soil input data.
o) L205 to 213, if the results are not described in this paper, this part can move to supplementary data.
p) L278 the measurement for leaf area index should be more detailed as it is an important variable. How thick is the canopy, and how did you correct the light coming from the side like the inter-row space? Leaf area index is better estimated if the radiation mainly coming from the top.
q) L299 “output data” do you mean the observation data or the simulation data.
r) L307 change the units for soil water potential.
s) L300 where is the results for model calibration?
t) L340 “thirteen days before” the days from April 5th to April 20th are 15 days.
u) Fig. 7 can move to supplementary as it is can be describe in one sentence in the main text.
v) L421 improve the clarity of the legend. E.g. Sensitivity of the berry sugar content (expressed in °Bx) to changes in air temperature. Berry sugar content is expressed as the value at the end of each simulation. ΔTair the difference between the input temperature and the actual temperature record.
w) L428 where does the 1.1 Bx comes from. 25.5 – 25 should be o.5 Bx.
x) Legend in Fig. 9, add maximum before LAI.
y) L487 “highest grid point” replace this expression into highest elevation site and the lowest grid point into lowest elevation site as well. It is really unclear what does the highest grid point means?
z) Fig. 14 is the yield trend agree with other long term yield records in Europe? What was the major yield reduction in 2003? Are there big temperature reduction due to volcano eruption? In 1982, 1991 and 1993 there were also big eruptions. So I feel the trends from 1980 to 2009 could be affected by those big events. After volcano explosion, you have a cool temperature and yield for the current year or the following year, then it gradually increase again.
aa) L531 does this agree with the observation? Are you records for this?
bb) Table 7 could move into supplementary.
cc) L606 rephrase this paragraph and move to the end of introduction to explain why this study was done.
dd) Add a section about what can we learn from this model and how this model can help us in improving vineyard management?
Author Response

(The authors gave the same response as above.)

Round 2
Reviewer 1 Report
The second revised version of the manuscript entitled “Description and Preliminary Simulations with the Italian Vineyard Integrated Numerical Model for Estimating Physiological Values (IVINE)”, authored by V. Andreoli, C. Cassardo, T. Laiacona and F. Spanna, (ref. agronomy-423553-v2) represents a great improvement from the two previous versions.
The authors accounted for all my comments, suggestions and corrections within the new version of their manuscript. Moreover, they answered satisfactorily to all my remarks. I appreciate their effort and thank them for this hard work.
The authors stated that some of my comments was a little bit offensive to them, I apologize for this. It was not my intention to offend anybody.
This is the third time I revise this manuscript and I only have some minor comments that can be corrected during the editing process since I feel that English can be improved. For instance, authors use “founded” in lines 1170, 1172, 1174, 1177, 1181 and 1183, when they clearly meant “found”. I am sure that this kind of mistakes will be corrected by the editorial office when proofreading the manuscript.
Author Response
Referee #1 asked to correct a grammar mistake in lines 568-581 (found instead of founded). We corrected it.
Reviewer 2 Report
The paper has been significantly improved in terms of method description and figures. however, I feel we may still want another round for improving the writings. For example, the description of the requirement to develop another grapevine model could be too harsh. The authors may want to reduce the key, and maybe say: Here we would like to build based on the previous method and develop a grapevine model that to study the effects of climate change on phenology and yield in northwestern Italian region Piedmont.
Author Response
Referee #2 asked to change a sentence in the Introduction (lines 113-118). We changed it.